# Satisfaction with Social Support and Quality of Life Among Portuguese Patients with Breast Cancer: Mediating Effects of Coping Styles—Cross-Sectional Study

**DOI:** 10.3390/healthcare13030297

**Published:** 2025-02-01

**Authors:** Joana Carreiro, Susana Cardoso, Pedro Teques, Andreia P. Teques, José Luís Pais-Ribeiro

**Affiliations:** 1Department of Social Sciences and Behavior, University of Maia, 4475-690 Maia, Portugal; 2Research Center in Sports Sciences, Health Sciences and Human Development (CIDESD), 5000-801 Vila Real, Portugal; cardoso@umaia.pt (S.C.); pteques@ipmaia.pt (P.T.); 3Laboratory of Neuropsychophysiology, Faculty of Psychology and Education Sciences, University of Porto, 4200-135 Porto, Portugal; 4N2i Polytechnic Institute of Maia, 4475-690 Maia, Portugal; apereirateques@ipmaia.pt; 5Faculty of Psychology and Education Sciences, University of Porto, 4200-135 Porto, Portugal; jlpr@fpce.up.pt; 6William James Center for Research, ISPA, 1149-041 Lisboa, Portugal

**Keywords:** breast cancer, coping styles, fighting spirit, mediation analysis, social support, structural equation modeling

## Abstract

**Background/Objectives**: The purpose of this study was to analyze the mediating effects of coping styles on the relationship between satisfaction with social support (SSS) and quality of life (QoL) in breast cancer patients. **Methods**: A cross-sectional structural equation modeling (SEM) approach was used to examine the mediating effects. The participants were 311 women who were breast cancer patients, aged between 27 and 86 years (M = 46.61, SD = 9.53). **Results**: SEM analyses showed that SSS was positively associated with fighting spirit (β = 0.31, *p* < 0.05), cognitive avoidance (β = 0.19, *p* > 0.05), and QoL (β = 0.21, *p* < 0.05), and negatively associated with helplessness–hopelessness (β = −0.28, *p* < 0.05). Furthermore, fighting spirit and helplessness–hopelessness showed significant relationships with QoL. Anxious preoccupation was negatively related to QoL. Furthermore, fighting spirit and helplessness–hopelessness showed significant relationships with quality of life (β = 0.18, *p* < 0.05; β = −0.15, *p* < 0.05, respectively). In addition, the mediation analysis revealed that coping styles (i.e., fighting spirit and helplessness–hopelessness) mediated the relationship between SSS and QoL. **Conclusions**: The findings suggest that perceptions of SSS from family and friends may promote perceived QoL via adaptive coping, such as fighting spirit and helplessness–hopelessness strategies to deal with the cancer experience.

## 1. Introduction

Breast cancer is one of the most frequently diagnosed cancers and the leading cause of cancer deaths among women worldwide [1,2]. In 2018, there were more than 2 million new cases of breast cancer worldwide, with 521,900 deaths attributed to this pathology, constituting the cancer disease in women with higher incidence, prevalence and mortality [2]. In 2022, an incidence of all cancers of 19,976,499 new cases was recorded worldwide, and the risk of developing cancer before the age of 75 is 18.5% for women. The three main cancers in females are breast, lung and colorectal cancer. In the same year, 9,743,832 deaths were recorded. In Portugal, the number of new cancer cases is 69,567, while for females’ breast cancer, 8954 new cases are recorded [2]. However, despite the incidence, the combination of an increase in available screening methods, earlier diagnosis, and the existence of better therapeutic methods has increased the number of survivors [3].

The diagnosis of breast cancer is typically recognized as an event that can induce stress in women, thus generating an imbalance in their well-being and quality of life that requires the mobilization of personal and social resources to deal with the situation [4,5,6,7,8].

In response to increased survival rates, there must be ongoing concern about the quality of medical and psychosocial care provided to promote well-being, quality of life (QoL), and adjustment to disease, with a positive impact on the oncologic rehabilitation process [1,5,7]. Several investigations have verified the existence of psychosocial factors external and internal to the individual that contribute to a better quality of life and adaptation to the disease, such as styles of coping with the disease [8,9,10,11,12,13,14,15,16] and perception of social support [5,7,11,15,17,18].

The ability of women to deal with stressful situations facilitates the process of adaptation to breast cancer [8,9,10], and confronting the disease state implies the adoption of a set of coping styles, namely fighting spirit (response pattern, characterized by an optimistic attitude, where the disease is seen as a challenge and is perceived as something that is under control), cognitive avoidance or denial (refers to the attempt not to think about the disease, characterized by the rejection of the diagnosis or the denial of its severity), fatalism (refers to a passive acceptance of the disease), hopelessness (a strategy characterized by pessimism and the absence of strategies to actively combat the disease, where the illness is seen as a loss) and anxiety–worry (characterized by persistent anxiety and compulsive searching for information, although interpreted pessimistically). These coping styles can be classified as active (fighting spirit and cognitive avoidance) or passive (fatalism, anxious worry, and hopelessness) [8,10,11,12,13,14,15].

Active coping styles are considered more effective when coping with breast cancer [9,14]. Active coping efforts are aimed at facing a problem directly and determining possible viable solutions to reduce the effect of a stressor. Instead, passive coping refers to behaviors that seek to escape the source of distress without confronting it [8,11,14]. However, coping with the disease is a dynamic and changeable process, depending on the diagnosis, prognosis and/or evolution of the disease, the therapeutic decisions, the personal characteristics of the woman and her self-esteem, and the impact of the disease on her priorities and life goals [6,8,19]. Thus, although active standards such as fighting spirit advocate for a better quality of life and better adaptation to the disease [9,14], emotion-focused coping seems to promote adaptation to extreme adversity, since avoiding thoughts, emotions, and memories seems to reduce distress in stressful situations [20,21].

Coping styles allow for more balanced emotional reactions and a reduction in the negative impact of the disease, contributing to a better quality of life and adaptation to the disease [4,8,9,10]. Conversely, failure to use coping styles can lead to significant psychosocial distress, which can affect an individual’s ability to cope effectively with the illness and result in poorer quality of life [4,8,14,15].

Social support refers to the support received (e.g., emotional, instrumental, or informative) or the sources of the support (e.g., friends, family, healthcare providers), i.e., the resources available to individuals in response to a request for help and assistance. It has the ability to alleviate distress in crisis situations, can inhibit the development of diseases and, when the individual is sick, plays a positive role in recovery from the disease [5,7,22,23,24]. Thus, social support proves to be a resource that may help individuals reinterpret their illness situation, reinforce their self-esteem, promote a sense of mastery and/or competence given the challenges that the disease imposes and also find positive meaning in the experience of the disease [5,7,22,23,24]. In this case, the existence of significant interpersonal relationships, which include various sources of social support, may contribute to the perception of the quality of life of cancer patients [5,7,9,15,23,25,26].

Thus, the experience of positive emotions and trajectories of resilience in situations that induce stress (i.e., diagnosis of cancer) may be associated with the use of active coping with the disease, such as fighting spirit and perception of good social support [5,6,7,8,14,15,20,23]. The resources available to deal with the disease, such as personal, psychological and social resources, seem to protect the patient from the invasion of cancer and its treatment, preserving their quality of life and, consequently, facilitating psychosocial adaptation to the disease and the oncological rehabilitation process [6,7,9,15,19,27].

Although the literature reveals the existence of associations between coping, social support and QoL, little is known about the mediating effects of coping styles on the relation between the satisfaction with social support and the quality of life of women diagnosed with breast cancer. In this context, the present study intends to expand the knowledge in this area through the testing of the simultaneous relations between the variables and the mediating effects of the coping styles.

### The Present Study

The purpose of the present study was to evaluate the mediating effects of coping on the relationship between the satisfaction with social support and the quality of life of women with breast cancer. Considering the literature that evidences associations between coping and social support [14,15,23,28] and quality of life [1,5,6,7,8,9,10,26], it is expected that the use of coping mediates the relationship between social support and quality of life. Specifically, we predicted that social support will be positively related to quality of life. Furthermore, active coping (i.e., fighting spirit and cognitive avoidance) will positively mediate the association between social support and quality of life, and passive coping (i.e., helplessness–hopelessness and anxious preoccupation) will negatively mediate this relationship [8,14,29] (Figure 1).

## 2. Materials and Methods

### 2.1. Design

This study used a descriptive quantitative methodology with a non-experimental cross-sectional design. The data collected were quantified and the evaluation was carried out at a single point in time [30].

### 2.2. Participants

The inclusion criteria for this study were women aged 18 or over, native speakers of Portuguese, and diagnosed with breast cancer. The exclusion criteria were those who fall into this category but cannot be included for some reason: a specific illness, pregnancy, and/or neurological disease.

The participants were 311 Portuguese women with breast cancer, aged between 27 and 86 years (mean = 46.61, SD = 9.53). Due to the objectives of the present study, the participants were women, aged 18 or over, had a diagnosis of breast cancer (38% were diagnosed between 2 and 5 years; 27% between 1 and 2 years; 26% for less than 6 months; 9% for more than 5 years), revealed knowledge and mastery of new information and communication technologies—internet and social networks, and finally, were members of an association, support group or self-help group on the social network Facebook.

### 2.3. Instruments

Satisfaction with social support. The Social Support Satisfaction Scale [31] evaluates the satisfaction with perceived social support in health contexts. It is a self-fulfillment scale, consisting of 15 items distributed by four factors related to satisfaction with the social support received, namely (1) Satisfaction with Friends (e.g., “I am satisfied with the amount of friends I have”; 5 items); (2) Satisfaction with Intimacy (e.g., “When I need to vent with I can someone easily meet friends with whom to do it”; 4 items); (3) Family Satisfaction (e.g., “I am satisfied with the way I relate to my family”; 3 items); and (4) Satisfaction with Social Activities (e.g., “I miss social activities that satisfy me”; 3 items). For each item, participants indicated the degree of agreement on a five-point Likert scale (from “Totally Agree” to “Totally Disagree”). In this study, the Cronbach’s alpha scores were 0.83 (Satisfaction with Friends), 0.64 (Satisfaction with Intimacy), 0.64 (Satisfaction with Family), and 0.85 (Satisfaction with Social Activities).

Coping. The Reduced Scale of Mental Adjustment to Cancer [32,33] assesses how individuals deal with the diagnosis and treatment of cancer. It is a multidimensional scale consisting of 29 items, distributed by five factors: (1) dejection/weakness (e.g., “I feel that life has no hope”; (2) anxious concern (e.g., “I am concerned about my illness”; 8 items); (3) fighting spirit (e.g., “I am determined to overcome my illness”; 4 items); (4) cognitive avoidance (e.g., “Not thinking about my illness helps me deal with it”; 4 items); and (5) fatalism (e.g., “At this moment I live one day at a time”; 5 items). Each item on this scale is a statement that describes the patient’s reactions to cancer in a four-point Likert-type response format (1—“It does not apply to me at all”; 4—“Applies entirely to me”). High values for the sub-scales of discomfort/weakness, anxious worry, and cognitive avoidance indicate low adaptive and negative coping levels, while high levels for the fighting spirit sub-scale indicate levels of adaptive and positive coping with cancer [33]. The Cronbach’s alpha values were 0.79 (discouragement/weakness), 0.79 (worrying concern, 0.72 (fighting spirit), 0.84 (cognitive avoidance), and 0.28 (fatalism). It was decided to exclude the dimension of fatalism from the hypothetical model under analysis due to the constant references in the literature to this dimension related to the limitations in its validity and reliability [32,34].

Quality of life. Quality of life (QoL) was measured through the EORTC QLQ-C30 [35], which contains a global QoL scale with seven response options, where 1 corresponds to a “bad” classification and 7 to an “optimal” classification in response to the question “How would you rate your overall quality of life over the last week?” High scores on this scale indicate better perceived QoL.

### 2.4. Procedures

This study was approved by the Department Council of the doctoral program of the Faculty of Medicine of the University of Salamanca. Initially, a survey was sent to the associations and support groups available in Portugal that are intended to support oncological patients and women with breast cancer available on the social network Facebook. After identifying the associations, support groups and self-help groups with work developed in the fight against cancer, the authorization was requested of those responsible for administration/management of the Facebook page to share the research protocol with its members. Once the research group was granted authorization to disclose the research in the respective online groups, the research protocol was made available, which included a presentation text about the study where they set out their purpose, objectives and procedures, and where the anonymity of data was assured. Completion of the questionnaire took approximately 20 min.

### 2.5. Data Analysis

A two-step robust maximum likelihood method of structural equation modeling approach was performed using IBM AMOS version 23 [36]. First, a confirmatory factor analysis was implemented to analyze the quality of the variable’s adjustment to its indicators. Second, the structural model was estimated to test the mediating effects, as recommended by Danner, Hagemann, and Fiedler [37]. Specifically, the satisfaction with social support variables (i.e., friends, family, and social activities) were conceptualized to have an indirect association with QoL, and the coping styles (i.e., fighting spirit, cognitive avoidance, helplessness–hopelessness, and anxious preoccupation) were considered as mediators. To avoid inflated measurement errors, which are a consequence of models with a high number of indicators, we followed the strategy of item parceling for each factor [38]. The bootstrap resampling procedure (1000 bootstrap samples) with 95% bias corrected confidence intervals (CIs) was used to test the significance of the direct and indirect effects. An indirect effect was considered significant (at ≤0.05) if its 95% CI did not include zero [39]. Four indexes were considered to estimate the adjustment of the model to the data: CFI and TLI > 0.90; RMSEA and SRMR < 0.08 [36].

## 3. Results

An initial analysis of the data exposed 1.2% of missing values with no patterns of missing data. Thus, the missing data were imputed using AMOS’s regression techniques. The Mardia coefficients (98.33) surpassed the expected values for multivariate normality (<5.00). Consequently, a Bollen–Stine bootstrap was applied for the following analysis [40]. Also, the collinearity was tested using the variance inflation factors (VIFs) between all the variables under study. The VIF values ranged between 1.24 (fighting spirit) and 2.36 (satisfaction with social support), showing adequate conditions to perform the regression analysis [36].

### 3.1. Descriptive Measures of Psychosocial Variables

The means, standard deviations, squared correlations, and average variance extracted among all the variables are presented in Table 1. Participants revealed a moderate level of satisfaction with social support (M = 2.48, SD = 0.74). Regarding the coping styles, participants perceived a mean of 2.99 (SD = 0.88) for helplessness–hopelessness, and moderate levels of fighting spirit (M = 2.01, SD = 0.88), cognitive avoidance (M = 1.95, SD = 0.70), and anxious preoccupation (M = 2.38, SD = 0.81). Participants also revealed moderate levels of QoL (M = 4.78, SD = 1.54).

The measurement model comprised five latent variables (i.e., satisfaction with social support, fighting spirit, cognitive avoidance, helplessness–hopelessness, and anxious preoccupation) and 11 observed variables. The measurement model showed a good fit to the data [χ2(44) = 62.23, *p* < 0.001, CFI = 0.98, SRMR = 0.05, RMSEA = 0.03 (CI = 0.02, 0.05)]. The correlation matrix showed significant correlations among all the study variables, with the exception of the relationships between cognitive avoidance and anxious preoccupation (r = −0.03, *p* > 0.05) and cognitive avoidance and quality of life (r = 0.06, *p* > 0.05).

### 3.2. The Mediation Model for the Relationships Between Satisfaction with Social Support, Coping Strategies and Quality of Life

The hypothesized mediation model, which included a direct path from satisfaction with social support to quality of life and four mediators (i.e., fighting spirit, cognitive avoidance, helplessness–hopelessness, and anxious preoccupation), showed a satisfactory fit to the data [*χ*2(57) = 112.36, *p* < 0.001, CFI = 0.96, SRMR = 0.04, RMSEA = 0.05 (CI = 0.05, 0.06)]. This model is presented in Figure 2.

As hypothesized, satisfaction with social support was associated with fighting spirit (β = 0.31, *p* < 0.05), helplessness–hopelessness (*β* = −0.28, *p* < 0.05), cognitive avoidance (*β* = 0.19, *p* > 0.05), and quality of life (*β* = 0.21, *p* < 0.05). Furthermore, fighting spirit and helplessness–hopelessness showed significant relationships with quality of life (*β* = 0.18, *p* < 0.05; *β* = −0.15, *p* < 0.05, respectively). Also, anxious preoccupation was negatively related to quality of life (*β* = −0.16, *p* < 0.05). Nevertheless, the standardized direct effect from satisfaction with social support to anxious preoccupation (*β* = 0.01, *p* > 0.05) was not statistically significant. Also, the standardized direct effect from cognitive avoidance to quality of life was not significant (*β* = 0.01, *p* > 0.05). Overall, satisfaction with social support and the coping style variables accounted for approximately 26% of the variance in quality of life (Figure 2).

Table 2 displays the results of the mediation analysis between satisfaction with social support, coping styles (i.e., fighting spirit, cognitive avoidance, helplessness–hopelessness, and anxious preoccupation) and quality of life. Satisfaction with social support showed significant indirect effects on quality of life via fighting spirit (*β* = 0.20, CI [0.03, 0.36]) and helplessness–hopelessness (*β* = −0.16, CI [−0.20, −0.01]).

## 4. Discussion

The purpose of this study was to examine the mediating effects of coping styles on the relationship between the social support (SS) and the quality of life (QoL) of women with breast cancer. Overall, the hypothesized associations were identified. Specifically, SS was associated with fighting spirit, helplessness–hopelessness, cognitive avoidance, and QoL. As high levels of perceived social support are associated with quality of life, consistently viewed as favorable for psychological well-being [7,14,15,18,19], individuals who perceive support coming from their most significant persons (e.g., family, friends) are likely to be determined to cope with cancer using active strategies (e.g., information search, adherence to treatment) and less predisposed to depression or isolation [14]. These findings suggest that the improvement of patients’ social support may promote patients’ active coping styles during the cancer process [5,8,14].

Consistent with previous studies, fighting spirit and helplessness–hopelessness showed significant relationships with QoL [8,13,14,15,29]. Specifically, the fighting spirit strategy is the most representative of the coping type of active coping, in which the cancer patient presents proactive behavior in the various stages of the disease, contributing to the increase in the QoL perception. However, the helplessness–hopelessness strategy is associated with the coping type of passive coping, related to cognitions and behaviors of a defeatist profile, in which the action of withdrawal and negativism intersects with the feeling of despondency apprehended, where the oncological disease is evaluated as a loss with repercussions in the various domains of the patient’s life. The strategies presented different relationships insofar as the results show that patients, in applying fighting spirit strategies, present a greater perception of QoL, and when using helplessness–hopelessness strategies, they perceive less QoL. Consistently, this association of the discouragement–weakness strategy with a negative impact and emotional state, as well as with the subject’s inactivity and disbelief in their course of illness [24], is consistently presented in the literature. In addition, anxious preoccupation was negatively related to QoL. Passive strategies such as anxious worry consistently present a worse adjustment to oncologic disease [12].

These data reinforce the relevance of the Nezu, Nezu and Cos intervention’s focus on anxiety symptomatology, in which problem-solving training guides patients from task to task, making them more active and self-efficacious [41]. The findings expand current knowledge by demonstrating the small-to-moderate mediating effects of coping (i.e., fighting spirit and helplessness–hopelessness) on the relationship between SS and QoL. The mediating effect of the coping process between the emotional response and the stressful situation is conceptually assumed, both through problem-focused coping and emotion-focused coping, emotionally balancing the stressor [21]. In the context of stressful experiences, evaluated as damage and threat, coping presents mediating effects between the triggered stress and its consequences, which are mostly positive for a long period of time, revealing adaptive effects [8,9,14,19]. Several studies have shown the mediating effects of coping , such as on the relationship between stress events and anxiety, depression, psychological distress, somatic complaints, neurophysiological outcomes in cancer, and adjustment to disease, more precisely through Reid-Arndt and Cox [42], Sarenmalm et al. [10] and Sharif and Khanekharab [9]. The results of the present study reveal that the use of fighting spirit strategies promotes a greater perception of social support, which results in an increase in the QoL.

These results reinforce the study by Yeung, Lu and Lin, which indicates that the coping strategy of self-efficacy includes the search for social support, which allows the subjects to regulate themselves cognitive-emotionally, thus promoting the psychological, social and spiritual development of QoL [27]. The same authors indicate that survivors of breast cancer who have more cognitive processes in their experience can perceive more support from their social environment. At this stage of the disease process, there are positive changes, such as an improvement and approximation in interpersonal relationships. In turn, it is in the social context and in close relationships with others that the coping process develops, so the perception of social support is a predictor of adjustment and QoL in cancer [5,7,15,18,28,43].

The limitations and guidelines for future research that should be considered are now mentioned. Primarily, the transversal design of this study does not allow us to make interpretations of the causal relationship between the variables. Longitudinal research that emphasizes processual effects over a certain period of a cancer patient’s life would add to the researchers’ understanding of how satisfaction with social support, coping, and QoL reciprocally impact each other. The psychometric weaknesses associated with the coping construct are evident in the use of the Mini Mac administration and may influence the results, as mentioned in other studies [44]. Future studies should consider other measures of coping that can assess other dimensions in which cancer patients may cope during the cancer continuum. In addition, evidence suggests that the type of cancer, stage of disease process, and life cycle phase may influence the outcomes, and this suggests the indicator variables to be controlled in future studies [16]. Moreover, cancer patients change their social relationships during the cancer process, e.g., enhancing affective relationships [5,7,15,43]. Future studies should understand the dyadic units in which this movement of relational approximation is more frequent and its benefits for the patient, their relatives and the state of the disease. As well, future research should address the type of family or social networks in which the relational changes do not take place, insofar as guiding characteristics are created for the early signaling of families/social groups at risk in oncological disease processes. The present study makes a contribution to the understanding of the relations between SS, coping and QoL in women with breast cancer. The sample collection method, on the one hand, made it possible to reach a considerable N and, on the other hand, reinforced the therapeutic effect of online social support communities, in which the sharing of life histories seems to suppress distress and isolation [22,45].

## 5. Conclusions

The purpose of this study was to analyze the mediating effects of coping styles on the relationship between SS and QoL in breast cancer patients. The findings suggest that the improvement of patients’ social support may promote patients’ active coping styles during the cancer process, such as the fighting spirit and helplessness–hopelessness strategies to deal with the cancer experience. Understanding the relationship between the variables that promote disease process adjustment reveals the relevance of developing the screening/assessment and monitoring of patient needs and of implementing intervention programs based on a psychosocial, patient-centered approach and which include/integrate elements of the most significant dyads of cancer patients. Additionally, the promotion of psychoeducational actions on the informal caregiver theme as a clinical practice highlights the benefits of sharing care in the informal social network, in which most of the family members are identified. The objective of guiding healthcare professionals to value and facilitate adjustment and efficiency in care delivery will result in a greater perception of social support in patients and their families.

## Figures and Tables

**Figure 1 healthcare-13-00297-f001:**
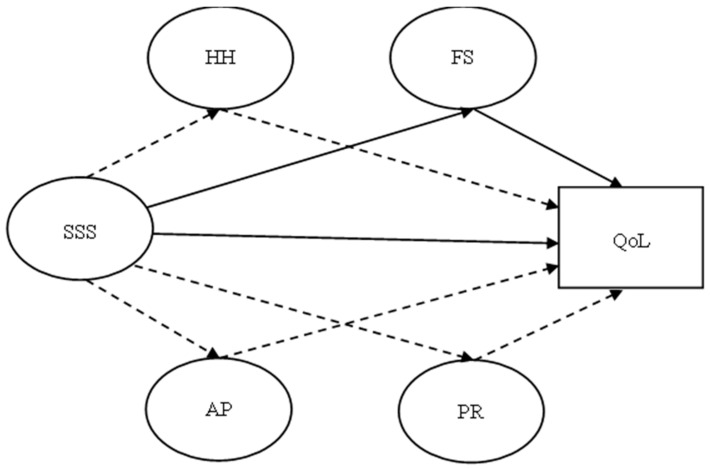
The expected mediation model for the relationships between satisfaction with social support (SSS), coping strategies (i.e., fighting spirit—FS, positive redefinition—PR, helplessness–hopelessness—HH, and anxious preoccupation—AP) and quality of life (QoL). Note: Positive paths in continuous lines and negative paths in dashed lines.

**Figure 2 healthcare-13-00297-f002:**
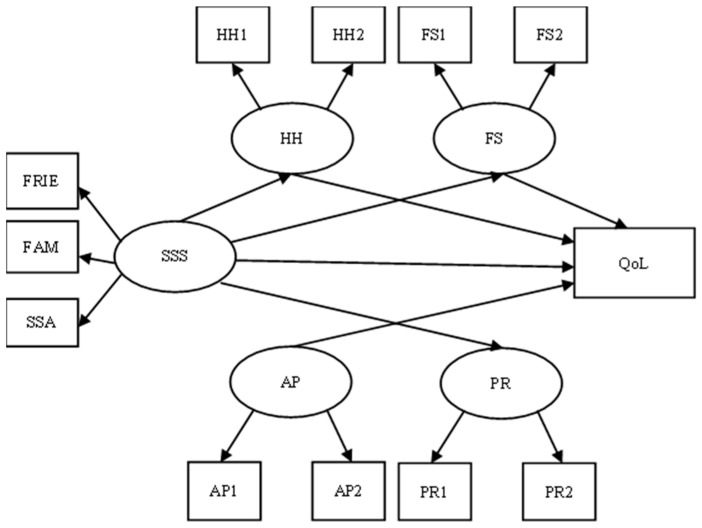
The mediation model for the relationships between satisfaction with social support (SSS), coping strategies (i.e., fighting spirit—FS, positive redefinition—PR, helplessness–hopelessness—HH, and anxious preoccupation—AP) and quality of life (QoL). Note: All the standardized path coefficients are significant at the 0.05 level; the value in bold is the coefficient of determination (R2).

**Table 1 healthcare-13-00297-t001:** Descriptive statistics and bivariate correlations for all the variables (n = 311).

Variables		1	2	3	4	5	6
1.	SSS	1					
2.	HH	−0.45 **	1				
3.	FS	0.46 **	−0.52 **	1			
4.	PR	0.18 *	0.22 **	0.05	1		
5.	AP	−0.15 *	0.54 **	−0.20 **	−0.04	1	
6.	QoL	0.32 **	−0.14 *	0.23 **	0.06	−0.22 *	1
	M	2.48	2.99	2.01	1.95	2.38	4.78
	DP	0.74	0.88	0.93	0.70	0.81	1.54
	Range	1–5	1–4	1–4	1–4	1–4	1–7

Note. SSS = satisfaction with social support, HH = helplessness–hopelessness, FS = fighting spirit, PR = positive redefinition, AP = anxious preoccupation, QoL = quality of life. * *p* < 0.05, ** *p* < 0.01.

**Table 2 healthcare-13-00297-t002:** Standardized indirect effects and confidence intervals.

Mediating Paths	Estimate	95% CI
		Lower
Satisfaction with social support → Fighting spirit → quality of life	0.20	0.03
Satisfaction with social support → Helplessness–hopelessness → quality of life	−0.16	−0.20

Note: Indirect effect is considered significant because its 95% CI does not include zero [39].

## Data Availability

The original contributions presented in this study are included in the article. Further inquiries can be directed to the corresponding authors.

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
