# Peer review of "Satisfaction with Social Support and Quality of Life Among Portuguese Patients with Breast Cancer: Mediating Effects of Coping Styles—Cross-Sectional Study"

_healthcare, 2025, doi:10.3390/healthcare13030297_

Round 1

Reviewer 1 Report

Comments and Suggestions for Authors

Comments

Thanks to the authors for the work on Mediation Effects of  Coping Styles in breast cancer patients. To make the work better, I have made some suggestions.

Abstract

Well written

Introduction

Line 37: I would prefer “female gender” or “women”

In the first paragraph of the introduction, I would suggest the authors expunge the details about cancer in men. Focus on cancer in women and narrow it down to breast, which is the subject of discussion.

Line 111-3 “The purpose of the present study was to evaluate the mediating effects of coping on 111 the relationship between satisfaction with social support and the quality of life of women 112 with breast cancer “ move this to this to the introduction.

Line 113-125: This should not be part of the introduction. It is too pre-emptive. It can form part of the hypothesis, which can best be placed in “Materials and methods” section.

Materials and methods

Line 133-4: Exclusion criteria are not direct opposite of inclusion criteria. There are those who fall into the study population but one or two issues that disqualify them e.g. other comorbidities. So, “The exclusion criteria were males, minors, those without a cancer diagnosis and were not receiving palliative treatment “ are not exclusion criteria in this study. The authors should state the real exclusion criteria in this study.

The study needed ethical approval as it involved human subjects 

Results

Line 214: Descriptive Measures  of what? Every title should be informative enough to guide the reader.

The results are poorly presented. The authors should present the results in a more logical and chronological manner

Discussion

Well discussed

Conclusion

Okay

Author Response

Dear Reviewer 1,

Thank you for your care and review of this manuscript. Please find below your response to each of your comments.

The changes made as a result of the review process are marked in blue in the manuscript.

Response to Reviewer1's comments:

Abstract

Comment: Well written

Response: Thank you for your comment.

Introduction

Comment: Line 37: I would prefer “female gender” or “women”

In the first paragraph of the introduction, I would suggest the authors expunge the details about cancer in men. Focus on cancer in women and narrow it down to breast, which is the subject of discussion.

Line 111-3 “The purpose of the present study was to evaluate the mediating effects of coping on 111 the relationship between satisfaction with social support and the quality of life of women 112 with breast cancer “ move this to this to the introduction.

Line 113-125: This should not be part of the introduction. It is too pre-emptive. It can form part of the hypothesis, which can best be placed in “Materials and methods” section.

Response: Thanks for the suggestion. We have made the suggested changes. Please see lines 37 and 40.

Materials and methods

Comment: Line 133-4: Exclusion criteria are not direct opposite of inclusion criteria. There are those who fall into the study population but one or two issues that disqualify them e.g. other comorbidities. So, “The exclusion criteria were males, minors, those without a cancer diagnosis and were not receiving palliative treatment “ are not exclusion criteria in this study. The authors should state the real exclusion criteria in this study.

The study needed ethical approval as it involved human subjects 

Response: Thank you for your comment. We would appreciate it if you could clarify your comment further, as we feel that the inclusion and exclusion criteria are clearly worded.

Regarding ethical approval, as we had already mentioned, approval for the study was not required according to the Internal Regulations of the Research Ethics Committee of the University of Salamanca.

Results

Comment: Line 214: Descriptive Measures  of what? Every title should be informative enough to guide the reader.

The results are poorly presented. The authors should present the results in a more logical and chronological manner.

Response: Thank you for your comment. We have made the proposed change to the title. Please see line 214.

With regard to the results, we believe that they have been described clearly and precisely, in accordance with good scientific writing practices. If you think any improvements are necessary, we welcome your suggestions.

Discussion

Comment: Well discussed

Response: Thank you for your comment.

Conclusion

Comment: Okay

Response: Thank you for your comment.

Reviewer 2 Report

Comments and Suggestions for Authors

SCOPE: the manuscript is in line with the thematic scope of the Healthcare.

TITLE: it should specifie the type of research  („Satisfaction with Social Support and Quality of Life among 2 Breast Cancer Portuguese Patients: Mediation Effects of 3 Coping Styles – cross-sectional study”)

ABSTRACT: the length and quality are correct.

KEYWORDS: number of keywords is acceptable.

INTRODUCTION: in the first sentence, more literature is needed to support this statement. Apart from that, this section is well developed and understandable.

MATERIALS AND METHODS: what hypotheses were tested? Transparent section, well-selected tools for measuring variables, the group of respondents could be larger considering the type of study (quantitative study), type of survey and wide scale of disease incidence.

RESULTS: this section has been written in a clear and concise manner. Maybe it would be worth plotting the numerical data on Figure no. 2? – this is just a suggestion.

DISSCUSSION: section developed correctly.

CONCLUSIONS: lines: 340-342 („Findings suggest that perceptions of SS from family and friends may promote perceived QoL via adaptive coping, such as fighting spirit and helplessness-hopelessness strategies to deal with cancer experience”) – is there data in the results section to confirm this assumption? Perhaps it would be worth expanding the Results section to include individual components of social suport (4 aspects)? This section is more postulative – there are no specific statements based on the results of this study.

REFERENCES: most publications are current and were published after 2015. The selection of publications could be more relevant and directly related to the problem discussed.

Author Response

Dear Reviewer 2,

Thank you for your care and review of this manuscript. Please find below your response to each of your comments.

The changes made as a result of the review process are marked in blue in the manuscript.

Response to Reviewer 2's comments:

Comment: SCOPE: the manuscript is in line with the thematic scope of the Healthcare.

Response: Thank you for your comment.

Comment: TITLE: it should specifie the type of research  („Satisfaction with Social Support and Quality of Life among 2 Breast Cancer Portuguese Patients: Mediation Effects of 3 Coping Styles – cross-sectional study”)

Response: Thank you for your suggestion. We have made the change you can see on line 4.

Comment: ABSTRACT: the length and quality are correct.

Response: Thank you for your comment.

Comment: KEYWORDS: number of keywords is acceptable.

Response: Thank you for your comment.

Comment: INTRODUCTION: in the first sentence, more literature is needed to support this statement. Apart from that, this section is well developed and understandable.

Response: Thank you for your comment. The IARC reference (2024) has been added to line 35.

Comment: MATERIALS AND METHODS: what hypotheses were tested? Transparent section, well-selected tools for measuring variables, the group of respondents could be larger considering the type of study (quantitative study), type of survey and wide scale of disease incidence.

Response: Thank you for your comment. Because the study was exploratory and the information in the scientific literature is not always consistent about the role and mediation of the various coping strategies, we decided not to formulate hypotheses so as not to bias the results and, therefore, produce results that could be a contribution to theory and could formulate hypotheses for future studies.

With regard to the sample size, in Portugal, the fact that we are investigating a vulnerable population makes access to it difficult, even so the number of participants is in line with the sample size estimate.

Comment: RESULTS: this section has been written in a clear and concise manner. Maybe it would be worth plotting the numerical data on Figure no. 2? – this is just a suggestion.

Response: Thanks for the suggestion.

Comment: DISSCUSSION: section developed correctly.

Comment: CONCLUSIONS: lines: 340-342 („Findings suggest that perceptions of SS from family and friends may promote perceived QoL via adaptive coping, such as fighting spirit and helplessness-hopelessness strategies to deal with cancer experience”) – is there data in the results section to confirm this assumption? Perhaps it would be worth expanding the Results section to include individual components of social suport (4 aspects)? This section is more postulative – there are no specific statements based on the results of this study.

Response: Thank you for your comment. We have reworded that statement. Please see lines 340-343.

Comment: REFERENCES: most publications are current and were published after 2015. The selection of publications could be more relevant and directly related to the problem discussed.

Response: Thank you for your comment. We believe that our references meet the objective of the study, design and are current.

Round 2

Reviewer 1 Report

Comments and Suggestions for Authors

Comments round two

Dear authors, thanks for making efforts to improve the manuscript.

Introduction

The authors did not make the changes suggested in the first round of the review. The suggestions in the first round were as follows:

1.      In the first paragraph of the introduction, I would suggest the authors expunge the details about cancer in men. Focus on cancer in women and narrow it down to breast, which is the subject of discussion.

2.      Line 111-3 “The purpose of the present study was to evaluate the mediating effects of coping on the relationship between satisfaction with social support and the quality of life of women with breast cancer “ move this to this to the introduction.

3.      Line 113-125: This should not be part of the introduction. It is too pre-emptive. It can form part of the hypothesis, which can best be placed in “Materials and methods” section.

Please refer to the first round of the review

Materials and methods

Dear authors, here is the clarification on inclusion and exclusion criteria

Since the inclusion criteria for this study were women aged 18 or over, native speakers of Portuguese, diagnosed with breast cancer. The exclusion criteria will be those who fall into this category but some reasons cannot be included. This reason may be a particular disease condition, pregnancy etc.

The ethical issue is accepted

Results

Line 217: The subtitle is fully informative. However, others have to be addressed in the same manner

Author Response

Dear reviewer 1,

Thank you for your comments.
1. We have made the changes suggested in the introduction regarding information on cancer in men. The changes have been marked in blue and can be found on lines 39-45.

2 & 3.  We sincerely appreciate your suggestion. However, following certain best practices in scientific writing, the objectives and hypotheses are typically presented at the end of the Introduction section and before the Methods section, as suggested by APA guidelines (7th edition). For this reason, we believe retaining the information as it is currently presented is appropriate. Thank you once again for your valuable feedback.

3. Thank you for your comment. We have made the suggested changes. Please see lines 133-136.

Results

Thank you for your comment.
We have made the suggested changes. Please see lines 234-235.

Reviewer 2 Report

Comments and Suggestions for Authors

In the reviewer's opinion, the corrections made (and there were few of them) are rather sufficient.

Author Response

Dear reviewer 2,
Thank you for your feedback, which has been a great contribution to our manuscript.